# Food Security in China: A Brief View of Rice Production in Recent 20 Years

**DOI:** 10.3390/foods11213324

**Published:** 2022-10-23

**Authors:** Ling Tang, Hamdulla Risalat, Rong Cao, Qinan Hu, Xiaoya Pan, Yaxin Hu, Guoyou Zhang

**Affiliations:** 1School of Management Science and Engineering, Nanjing University of Information Science & Technology, Nanjing 210044, China; 2Key Laboratory of Agrometeorology of Jiangsu Province, School of Applied Meteorology, Nanjing University of Information Science & Technology, Nanjing 210044, China; 3Changwang School of Honors, Nanjing University of Information Science & Technology, Nanjing 210044, China

**Keywords:** grain yield, planting area, agricultural resources, food security, rice cultivars

## Abstract

Rice production affects the food security and socioeconomic status of over half the world’s population. Rice-producing countries, however, are facing population growth, reduction in rice planting area, and global change. Understanding the trends of rice production and major determinants is key to regulating rice production. We thus analyzed the trends of rice production and related determinants in China from 2001 to 2021, revealing that the annual rice production (TRP) has risen steadily (r = 0.929, *p* < 0.0001) in recent 20 years. TRP in 2021 was 19.9% higher than that in 2001, which was primarily achieved by the increment of middle rice production (MRP). MRP increased by 46.2% from 2000 to 2018, and grain yield per unit area (GPA) was the largest in middle rice. The enhancement of GPAs is significantly correlated with the consumption of agricultural resources and the number of released rice cultivars, but variations exist. TRP and GPA vary in different provinces; Hunan (25 ± 2 megatons) and Xinjiang (8364 ± 806 kg/hectare) show the largest values, respectively. TRP could be further increased by 13.8% by improving MRP. The results suggest that rice production in China has a large potential to be further improved through regulations.

## 1. Introduction

Rice (*Oryza sativa* L.), one of the most important crops, provides a stable source of food for more than half of the world’s population. In 2019, there were 501 megatons of milled rice produced in the world, and mean consumption was around 179 g per capita per day, 19.3% higher than that in 2000. The world production of milled rice was projected to rise to 508.4 megatons in 2020 [1]. However, according to the latest report on “Repurposing food and agricultural policies to make healthy diets more affordable”, up to 828 million people were affected by hunger in 2021 [2]. Because rice is the main staple food in more than 100 countries in the world, enhancement in rice production is crucial to ending hunger [2], especially in Asia, Sub-Saharan Africa, and South America regions. However, the global consumption and production of milled rice were predicted to be 519.9 and 503.6 megatons in 2022/23 [3]; thus, filling the gap between rice consumption and production is crucial. It is, therefore, essential to clarify the trends of rice production, especially among the main rice producers.

The world’s biggest rice producer is China, where the earliest domesticated rice can be traced back as early as 9000 years ago in the Yangtze Valley [4,5]. Currently, nearly every province in China plants rice; altogether, China produces more than one-fourth of the world’s rice annually [1,2,3]. In China, rice provides energy and nutrition for about 65% of the people [6]. Changes in rice production affect not only food security, but also the socioeconomic status of a large population. As ensuring the absolute security of staple food is the top issue [6], effective regulations on maintaining and improving rice production are important to ensure food security [7,8,9,10]. A detailed analysis of the trends in rice production in the recent 20 years is lacking.

According to the white paper on Food security in China (The State Council Information Office of the People’s Republic of China, 2019), rice production per unit area (GPA) reached 6916.9 kg/hectare by 2017, which was 11.3% higher than that in 1996. Whether the GPA continued to rise in recent years and how the other factors that determine rice production changed need to be revealed. According to the planting date, rice production could be classified into three types: early rice, middle rice, and late rice. Significant spatiotemporal distributions exist in the rice production in China [11]. Whether the rise of rice production was achieved by the increase in GPA, the planting area or other factors are not yet clarified. The variations of the change in rice production in different rice types may reveal important information for agronomic scientists and policymakers to further improve rice production. In order to investigate the trend of rice production in recent years, the reasonable factors that regulate rice production, and the potential to further improve rice production, we thus conducted this study.

## 2. Materials and Methods

In China, according to the planting date, rice production could be classified into three types: early rice, middle rice, and late rice (Table 1); for example, rice production in Wuhan, China, could be conducted from April to July (early rice), May to October (middle rice) and July to November (late rice) [12,13,14].

### 2.1. Data Collection

The data on rice production and consumption of agricultural resources were obtained from the statistical database of the National Bureau of Statistics of China [15]. According to the availability of the database, the rice production area of the whole country (mainland) was obtained from 2001 to 2021. The national rice production (TRP) and early rice production (ERP) data were obtained from 2001 to 2021, middle (MRP) and late rice production (LRP) data were from 2001 to 2018, the agricultural resource consumption data (effective irrigation area, agricultural fertilizer, nitrogen fertilizer, phosphorus fertilizer, potassium fertilizer and compound fertilizer, pesticide) that were used by all of the agricultural productions in the whole country were obtained. Data for annually released new rice cultivars by the country and provinces were extracted from published articles [16,17,18].

### 2.2. Statistical Analysis

Rice production, rice planting area, and grain yield per unit planting area in total, early, middle, and late rice were plotted against years. In order to see the correlations between traits and years and the trends of traits against years, Pearson correlation, pairwise correlation, ellipse method, and polynomial regression methods were adopted [19,20,21]. The goodness of fit is shown by R-squared and r values; the significance of the slope is shown by the *p*-values. Multiple correlations were used to analyze the relationships among the factors. The statistical analysis was mainly conducted by JMP-Pro ver. 16 (SAS Institute, Cary, NC, USA).

## 3. Results

### 3.1. Trends of Rice Production Are Different Depending on Rice Types, While TRP Keeps Rising

From 178 megatonnes in 2001 to 213 megatonnes in 2021, the annual TRP in China increased by 19.9% (Figure 1), and a significant linear correlation between TRP and year was found (r = 0.929, *p* < 0.0001). On the contrary, ERP was declining (r = −0.587, *p* = 0.0052), and ERP was 18% smaller in 2021 than that in 2001. The MRP in 2018 was 46.2% larger (Table 2) than that in 2001 (r = 0.984, *p* < 0.0001). LRP showed a decreasing trend, while the effect by year was not significant (*p* = 0.2181).

### 3.2. TPA and MPA Are Increasing, But EPA and LPA Are Decreasing

From 2001 to 2021, the annual TPA kept rising (R^2^ = 0.545, *p* = 0.0001), with a mean value of 29,635 kilohectares per year (Figure 2), in which the proportion of MPA was more than twice EPA and LPA. The trends of EPA, MPA, and LPA against year showed significant linear correlations; only MPA was increasing (R^2^ = 0.971, *p* < 0.0001). In comparison with 2001, TPA in 2021 increased by 3.8% (Table 2).

### 3.3. Grain Yield per Unit Planting Area in Rice Keeps Rising

The EGPA, MGPA, LGPA, and GPA show significant linear correlations with the year (Figure 3). The largest values of GPA, EGPA, MGPA, and LGPA were 7059 (kg/hectare) in 2019, 5967 (kg/hectare) in 2018, 7559 (kg/hectare) in 2018, and 5992 (kg/hectare) in 2015, respectively (Table 2).

### 3.4. Rice Production and Its Correlations with Planting Area and Grain Yield per Unit Area

In Figure 4, the multicorrelations among TRP and ERP, MRP, LRP, TPA, EPA, MPA, LPA, GPA, EGPA, MGPA, and LGPA are shown. TRP is most positively correlated with MRP (r = 0.98, *p* < 0.0001), MPA (r = 0.98, *p* < 0.0001), GPA (r = 0.97, *p* < 0.0001), and MGPA (r = 0.96, *p* < 0.0001).

### 3.5. Correlations of the GPA in Rice with the Consumption of Agricultural Resources

GPA, EGPA, MGPA, and LGPA were positively correlated with EIA, fertilization, and pesticide (Figure 5 and Figure 6). For GPA, EGPA, and MGPA, the coefficients (R^2^) were largest with EIA, followed by fertilization and pesticide. For LGPA, the rank was opposite, in which the largest coefficient was with pesticide, followed by fertilization and EIA.

Correlations of GPA, EGPA, MGPA, and LGPA with different kinds of fertilizer are shown in Figure 7. Nitrogen was not significantly correlated with GPA, EGPA, MGPA, or LGPA. Phosphate positively correlated with MGPA (R^2^ = 0.396, *p* = 0.0068) and LGPA (R^2^ = 0.571, *p* = 0.0004). Potassium positively correlated with GPA, EGPA, MGPA, and LGPA; the coefficient was largest in LGPA (R^2^ = 0.87). Compound fertilizer positively correlated with GPA, EGPA, MGPA, and LGPA; the coefficient was largest in GPA (R^2^ = 0.952).

### 3.6. Relations of Rice Yield to the Number of Released Rice Cultivars

Pearson correlations were analyzed to see the relationship between TRP and GPA in total, early, middle, and late rice with the number of annually released cultivars (Table 3). GPA, EGPA, MGPA, and LGPA are significantly correlated with the number of released rice cultivars. The largest coefficient was found between MRP with the number of new released rice cultivars by country (R = 0.595). The GAP was significantly correlated with the number of released rice cultivars by central government, local provinces, and country, with rank of whole country (R = 0.571) > local provinces (R = 0.557) > central government (R = 0.473). ERP and LRP negatively correlated with the number of new released cultivars.

### 3.7. Rice Production in Different Provinces

The mean TRP in China (mainland) in the recent 20 years was 197 megatons (Figure 8), in which 12.8% was contributed by Hunan (25 ± 2 megatons), followed by Heilongjiang (10.7%, 21 ± 7 megatons), Jiangxi (9.7%, 19 ± 2 megatons), Jiangsu (9.2%, 18 ± 1 megatons), Hubei (8.4%, 17 ± 2 megatons), Sichuan (7.4%, 15 ± 0.4 megatons), Anhui (7.3%, 14 ± 2 megatons), Guangxi (5.5%, 11 ± 1 megatons), Guangdong (5.4%, 11 ± 1 megatons) and the other provinces; there was no data on rice production in Qinghai yet.

In Figure 9, the GPAs in different provinces are shown. As there was no data on rice production in Qinghai, the smallest GPA was in that province. The smallest GPA in the province that has rice production data was Hainan (4685 ± 469 kg/hectare), which was 28.6% lower than the GPA in the whole country (6561 ± 310 kg/hectare). GPAs in Shanxi, Tibet, Guangxi, Guangdong, Guizhou, Jiangxi, Fujian, Yunnan, Anhui, Hunan, Hebei, and Beijing were lower than that in the whole country. Shaanxi, Inner Mongolia, Heilongjiang, Zhejiang, Chongqing, Gansu, Henan, Sichuan, Jilin, Hubei, Tianjin, Liaoning, Shandong, Shanghai, Jiangsu, Ningxia, and Xinjiang have larger GPAs than the whole country, GPAs in Xinjiang, Ningxia, Jiangsu, Shanghai, and Shandong are 27.5%, 26.8%, 26.0%, 25.9%, and 24.1% larger than that in the whole country.

### 3.8. Potentials for the Increment of Rice Production

In Figure 10, TRP potentials by improving MPA and EGPA are shown. If the EPA, LPA, and both together are transferred to MPA, calculated with MGPAs in different years accordingly, TRPs could be increased from 197 megatons to 203 megatons (TRP plus), 204 megatons (TRP plus 2), and 213 megatons (TRP plus 3). TRP plus 4 and TRP plus 5 are TRP potentials calculated by TPA with mean EGPA (7177 kg/hectare) and max EGPA (7559 kg/hectare) in the recent 20 years, respectively. TRP plus 4 is 8% larger than TRP, and TRP plus 5 is 13.8% larger than TRP.

## 4. Discussion

### 4.1. Why Does Rice Production in China in the Recent 20 Years Keep Rising?

Through this study, the trend of rice production in the recent 20 years and the related factors were analyzed; the results show that the annual rice production in China has been rising steadily (Figure 1). Current TRP was 19.9% higher than that 20 years ago, while the changes in TPA were small (Figure 2). GPA increased by 14.5% (Figure 3). These results suggest that the rise of rice production in China was mostly achieved by the enhancement of grain yield per unit area [22,23]. GPA was decided by rice cultivars, environment [24,25,26], field management, and the consumption of agricultural resources, including water, fertilizer, and pesticide change in these factors affect TRP. Because the data on agricultural resources consumption in recent years are not specific for rice, but for all agricultural productions in the country, uncertainties exist in this preliminary analysis, which, in turn, suggest future studies are needed.

#### 4.1.1. Enhancing GAP by Breeding New Cultivars

The newly released cultivars, especially the hybrid cultivars, have been found to have larger yield potentials [27,28]. There were 11,100 rice cultivars released by the country (1641) and provinces (9459) from 2000 to 2019 [29,30], during which agronomic traits and resistance to biotic and abiotic stresses were improved to ensure higher yield potentials. Both the TRP and GPA were significantly correlated with the number of annually released new rice cultivars (Table 3), indicating that the rise of rice production in China is also contributed by the release of new rice cultivars [31,32]. The R coefficient was largest in MRP, suggesting that middle rice production benefited most from the released new rice cultivars. Specific rice type or cultivar contributions to rice production should be analyzed when the released rice cultivar planting area data are available [27,28,31], although there are numerous new cultivars with various agronomic traits and ecotypes that will be released [33,34,35].

#### 4.1.2. GAP and Consumption of Agricultural Resources

Among the resources used in the whole agricultural production in China, we found that in the recent 20 years (Figure 5 and Figure 6), the rise of GPA was mostly correlated with the increase in water consumption (EIA, R^2^ = 0.950), followed by fertilization (R^2^ = 0.755) and pesticide (R^2^ = 0.739) consumption. The relations between EGPA, MGPA, and agricultural resource consumption are similar to GPA. However, LGPA correlates most with pesticide, followed by fertilization and EIA (Figure 5). Variations in the consumption of agricultural resources among early, middle, and late rice should be mostly induced by the differences among rice types. The differences among early, middle, and late rice are not only the planting and harvesting dates; the related climate factors and agronomic performance of different types of rice are also different [36]. Studies find that climatic factors account for 74.585% of the yield variation in late rice [37], while others find that transplanting date [38] and rice cultivars [39] are more important than climate. Climate and environmental variations affect plant diseases, such as rice blast, which is more severe in late rice in China [40,41,42,43]; this is probably the reason why GPA in late rice correlates the most with pesticide. Investigations on pesticides and the other agricultural resources used in different rice types and the underlying agronomic and climatic mechanisms should be conducted in the future.

#### 4.1.3. GAP in Different Provinces

The mean GPA in the whole country was 6561 kg/hectare (Figure 9), which was smaller than the GPAs in 17 provinces. Xinjiang had the largest GPA (8364 ± 806 kg/hectare), followed by Ningxia (8318 ± 256 kg/hectare), Jiangsu (8264 ± 373 kg/hectare), Shanghai (8261 ± 299 kg/hectare), Shandong (8139 ± 517 kg/hectare), and other provinces. Among the top five GPA provinces, only Jiangsu had a large TRP (Figure 8), suggesting that improving TPA in Ningxia, Shanghai, Shandong, and other provinces may further improve TRP.

### 4.2. Could Rice Production in China Continue to Rise in the Future?

TRP and MRP are increasing (Figure 1), but ERP and LRP are decreasing; thus, the rising TRP is mainly contributed by MRP. Middle rice is planted in almost all provinces in mainland China (Table 1), except Guangdong, Hainan, and Qinghai. Further increases of TRP by MRP could be achieved by enlarging the planting area of middle rice (MPA), enhancing grain yield per unit area in middle rice (EGPA), and increases in both MPA and EGPA. We calculated the TRP potentials by the current level of mean and max EGPA with TPA (Figure 10), revealing that TRP could be further increased by 13.8% through improving MRP. TRP and GPA in different provinces suggest that there is a large potential to further increase TRP in the whole country through regulations on the rice planting area, which remains to be clarified.

The changes in GPA in early, middle, and late rice are significantly correlated with the consumption of agricultural resources (Figure 7), except for nitrogen fertilizer (Figure 5), which may be caused by the regulation on the use of nitrogen fertilizer and improvement on the use efficiency owing to the need for environmental protection [44,45,46]. Although some strategies, such as alternate wetting and fixed-time adjustable-dose nitrogen management, could enhance water and nitrogen use efficiency [47,48], the future application and promotion of the strategies are needed. Moreover, as early, middle, and late rice are planted in different types of climates, soil, weather, and crop management systems, based on the national regulation and strategies, adjustments on the field management in local areas which suits the local climate and conditions would have the potential to further increase rice production.

## 5. Conclusions

In conclusion, rice production in the recent 20 years in China has been rising steadily, which was mostly achieved by the enhancement of the GPA. Among the three types of rice, the production of middle rice and its ratio to total rice production are rising; however, the production and ratio of early and late rice are decreasing. As middle rice has a larger GPA than early and late rice, the regulation of rice types and promotion of planting middle rice could further improve rice production in China. Our results showed that GAPs in early, middle, late, and total rice were significantly correlated with the consumption of agricultural resources. Because of the low availability of databases for agricultural consumption for all agricultural production in China, further studies on agricultural rice consumption should be conducted. The number of released rice cultivars contributed to the rise of rice production in the whole country, and further studies on the contribution of different cultivars to rice production should be conducted so that scientists in breeding and agronomic science and policymakers can improve rice production regulations.

## Figures and Tables

**Figure 1 foods-11-03324-f001:**
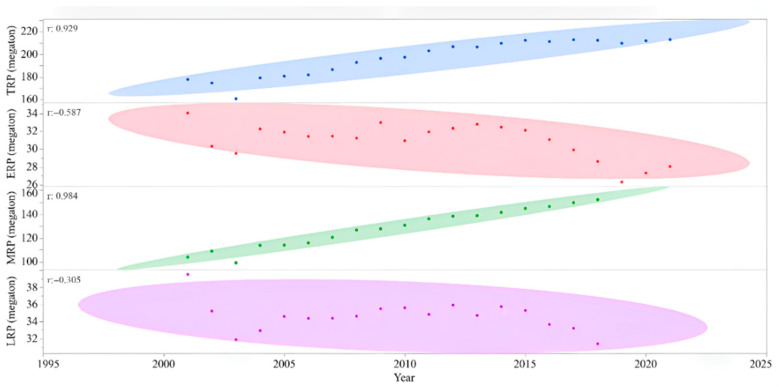
Total rice production (TRP), early rice production (ERP), middle rice production (MRP), and late rice production (LRP) in the recent 20 years. Correlations (r-value) are from ellipse method.

**Figure 2 foods-11-03324-f002:**
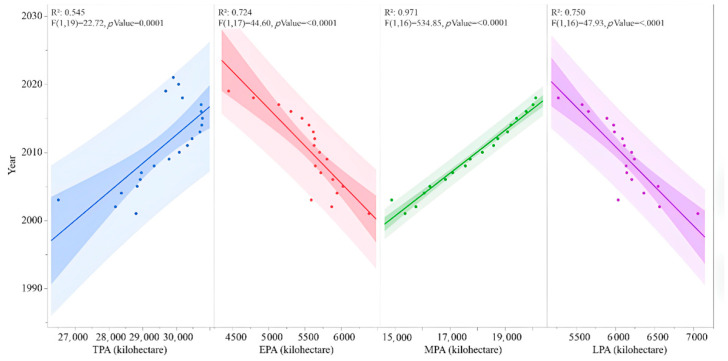
Total rice planting area (TPA), early rice planting area (EPA), middle rice planting area (MPA), and late rice planting area (LPA) in the recent 20 years. In order to fit the data and present the trends of the traits, polynomial regression, which fits a nonlinear relationship between the value of x and the corresponding conditional mean of y, was adopted. Predictions, R-squares, and results of F-test are from the polynomial regression model.

**Figure 3 foods-11-03324-f003:**
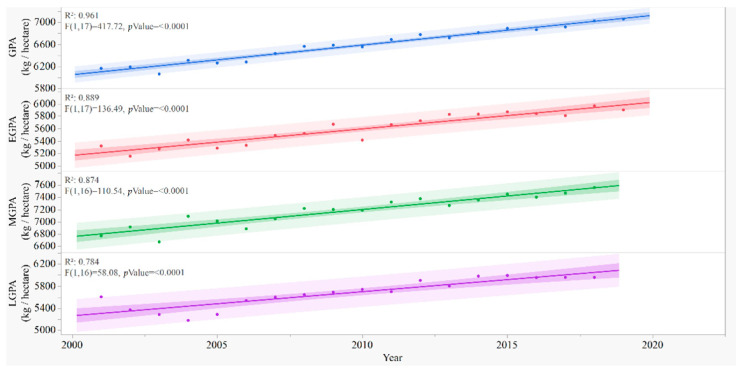
Grain yield per unit area (GPA) in total, early (EGPA), middle (MGPA), and late rice (LGPA) in the recent 20 years. Predictions, and R-squares, and results of F-test are from the polynomial regression model.

**Figure 4 foods-11-03324-f004:**
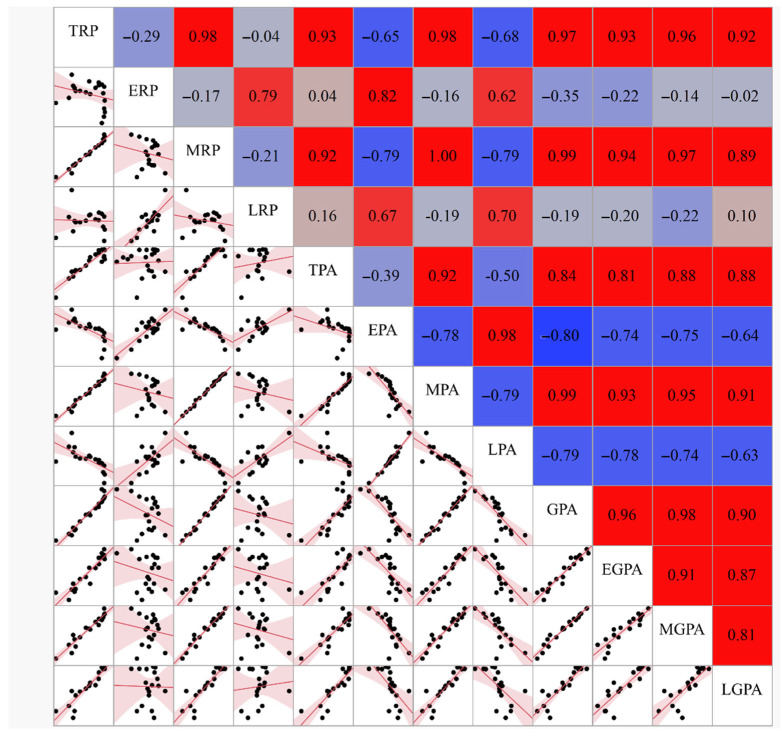
Multicorrelations among the rice production, planting area, and grain yield per unit planting area in early, middle, late, and total rice. The correlations are estimated by the pairwise correlation method.

**Figure 5 foods-11-03324-f005:**
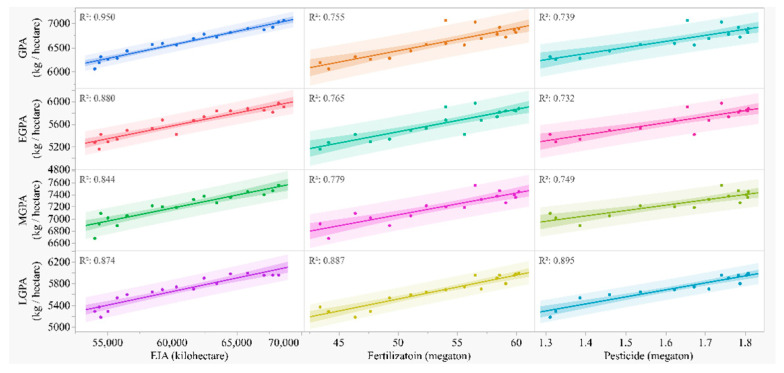
Grain yield per unit area (GPA) in total, early (EGPA), middle (MGPA), and late rice (LGPA) in the recent 20 years against effective irrigation area (EIA), fertilization, and pesticide used by the whole agricultural production in the country. Predictions and R-squares are from the polynomial regression model.

**Figure 6 foods-11-03324-f006:**
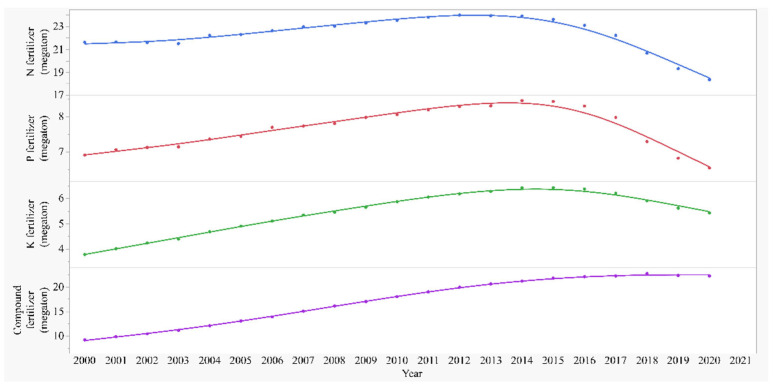
Different types of fertilizers (nitrogen, phosphate, potassium, and compound) used by the whole agricultural production in the country in the recent 20 years.

**Figure 7 foods-11-03324-f007:**
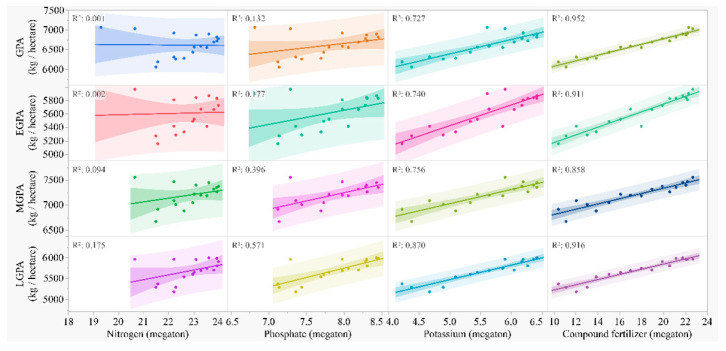
Grain yield per unit area (GPA) in total, early (EGPA), middle (MGPA), and late rice (LGPA) in the recent 20 years against different types of fertilizer (nitrogen, phosphate, potassium, and compound) used by the whole agricultural production in the country. Prediction andR-squares are from the polynomial regression model.

**Figure 8 foods-11-03324-f008:**
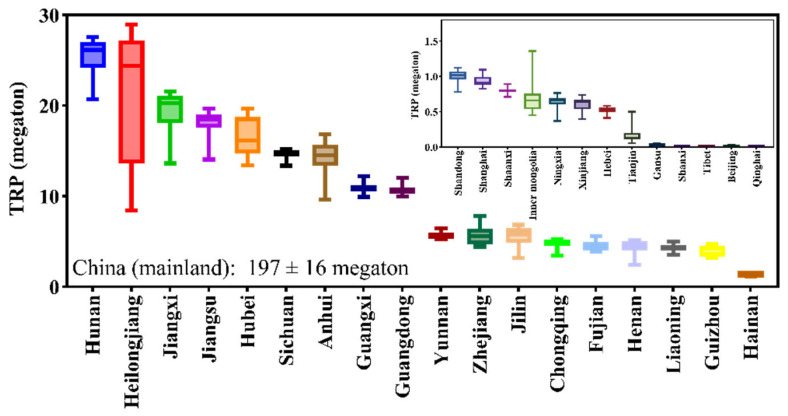
Rice production in different provinces in the recent 20 years. Error bars are the maximum and minimum values.

**Figure 9 foods-11-03324-f009:**
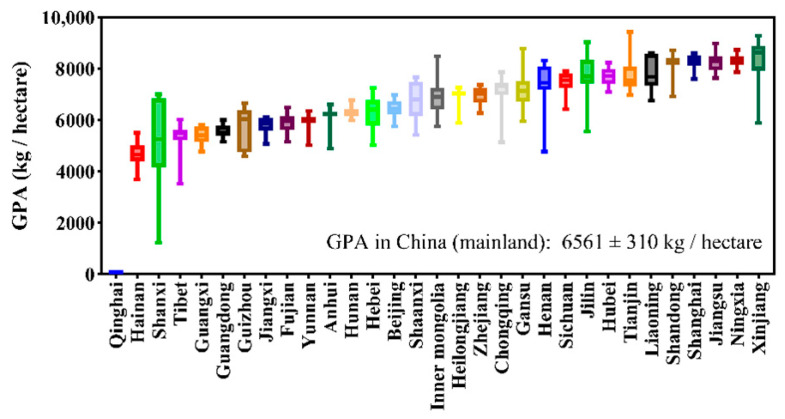
Grain yield per unit planting area in rice in different provinces in the recent 20 years. Error bars are the maximum and minimum values.

**Figure 10 foods-11-03324-f010:**
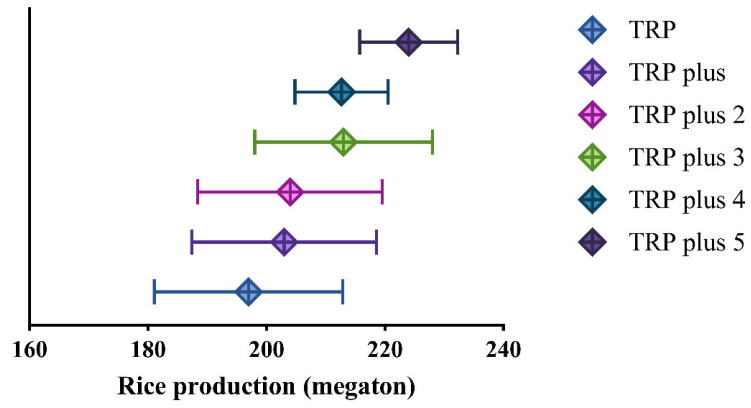
Potential TRPs. TRP plus: transferring early rice planting area to middle rice; TRP plus 2: transferring late rice planting area to middle rice; TRP plus 3: transferring early and late rice planting area to middle rice; TRP plus 4: transferring TRP plus 3 with mean EGPA; TRP plus 5: transferring TRP plus 3 with max EGPA. Error bars are standard deviations.

**Table 1 foods-11-03324-t001:** Types of rice production in different provinces in China (mainland).

Provinces	Early Rice	Middle Rice	Late Rice
Beijing		●	
Chongqing		●	
Gansu		●	
Guizhou		●	
Hebei		●	
Heilongjiang		●	
Henan		●	
Inner Mongolia		●	
Jiangsu		●	
Jilin		●	
Liaoning		●	
Ningxia		●	
Shaanxi		●	
Shandong		●	
Shanghai		●	
Shanxi		●	
Sichuan		●	
Tianjin		●	
Tibet		●	
Xinjiang		●	
Guangdong	●		●
Hainan	●		●
Anhui	●	●	●
Fujian	●	●	●
Guangxi	●	●	●
Hubei	●	●	●
Hunan	●	●	●
Jiangxi	●	●	●
Yunnan	●	●	●
Zhejiang	●	●	●
Qinghai	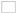	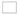	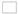

**Table 2 foods-11-03324-t002:** Mean, max, min, and the changes in rice production, planting area, and grain yield per unit planting area in the recent 20 years.

	TRP	ERP	MRP	LRP	TPA	EPA	MPA	LPA	GPA	EGPA	MGPA	LGPA
Year	***	**	***	ns	***	***	***	***	***	***	***	***
	megaton				kilohectare				kg/hectare			
Mean	197	31	128	35	29635	5589	17823	6116	6561	5580	7177	5677
Max	213	34	152	40	30784	6388	20125	7054	7059	5967	7559	5992
Min	161	26	99	31	26508	4450	14881	5273	6061	5158	6671	5179
Changes (%)												
Latest vs. 2001	19.9	−17.6	46.2	−20.5	3.8	−30.3	30.9	−25.2	14.5	10.9	11.7	6.3
Latest vs. mean	8.1	10.2	18.6	14.1	3.9	14.3	12.9	15.3	7.6	6.9	5.3	5.6
Latest vs. min	32.5	6.7	53.2	0.0	12.9	0.0	35.2	0.0	16.5	14.4	13.3	15.0
Max vs. mean	8.1	10.2	18.6	14.1	3.9	14.3	12.9	15.3	7.6	6.9	5.3	5.6
Min vs. mean	−18.4	−14.9	−22.6	−9.3	−10.6	−20.4	−16.5	−13.8	−7.6	−7.6	−7.1	−8.8

TRP, ERP, MRP, LRP, TPA, EPA, MPA, LPA, GPA, EGPA, MGPA, LGPA stand for total rice production, early rice production, middle rice production, late rice production, total rice planting area, early rice planting area, middle rice planting area, late rice planting area, grain yield per unit area, grain yield per unit area in early rice, grain yield per unit area in middle rice, and grain yield per unit area in late rice. Significance (*p* > 0.05, ns; *p* < 0.01, **; *p* < 0.001, ***).

**Table 3 foods-11-03324-t003:** Pearson correlation coefficients (R) between rice yield and the annual released number of new rice cultivars (ARC).

	GPA	EGPA	MGPA	LGPA	TRP	ERP	MRP	LRP
From central government	0.473 *	0.437 ^†^	0.423 ^†^	0.348	0.382 ^†^	−0.613 **	0.472 *	−0.612 **
From local provinces	0.557 *	0.420 ^†^	0.526 *	0.474 *	0.486 *	−0.511 *	0.592 **	−0.480 *
From country	0.571 *	0.461 *	0.531 *	0.465 *	0.465 *	−0.591 **	0.595 **	−0.568 *

GPA, EGPA, MGPA, LGPA, TRP, ERP, MRP, and LRP stand for grain yield per unit area, GPA in early rice, GPA in middle rice, GPA in late rice, total rice production, TRP in early rice, TRP in middle rice, and TRP in late rice. Data for ERP, MRP, LRP, and ARC are from 2000 to 2018, TRP and ARC are from 2000 to 2019. *, **, and ^†^ after the number indicates significant level at *p* < 0.05, *p* < 0.01, and *p* < 0.1, respectively.

## Data Availability

The data presented in this study are available on request from the corresponding author.

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
