# Peer review of "Food Security in China: A Brief View of Rice Production in Recent 20 Years"

_foods, 2022, doi:10.3390/foods11213324_

Round 1

Reviewer 1 Report

-         The abstract requires improvement by adding more explanation regarding the purpose of the study, methods and more results should be present. It should be more informative.

 -        The authors do not make a complete revision of the background of this topic

-        The structure of the paper is well established. The methods are explained in the Introduction part make it confusing. In the introduction, the topic, problem statement, and research gaps should be explained more in detail.

-         the problem and research gaps on that and then present the main contribution of the work.

-       Also, the author should provide a section for literature review and discuss more works related to the given task.

-        The author should provide a good systematic review, explain the limitation of the methods, and then provide the added values of their own proposed correction method

-        In the methodology part the Author didn’t explained the methods of statistical analysis of their work

-        Could the Author represent the data of fertilization (N,P,K), Descripe statistical methods

-         the Author didn’t makes a list of abbreviation in the end of the paper it will be better if its inserted in the text also for table 2 should define in the table or in the header of the table

-        The Authors showed different type of statistical methods but there is no any details in Methodology section could they explain how they anlyse the data

Author Response

Dear reviwer

Thanks very much for your time and valuable comments on our manuscript, which would definitely improve the presentation of our study. We have revised the manuscript according to your comments, details are as follows:

  1. We revised the abstract, try to introduce more information within the limitation of 200 words.
  2. The introduction part was revised: study background, the topic, problem statement, and research gaps as well as the purpose of the study were clarified.
  3. Added details for the statistical work in Methodology section.
  4. N, P, K and compound fertilizers consumptions data were shown in a Figure (Figure 6 in revised manuscript).
  5. A list of abbreviation in the end of the paper was added.

    Thanks very much for your time and comments.

    On behalf of all coauthors 

    Guoyou ZHANG, Ph.D, Associate Professor

    School of Applied Meteorology,

    Nanjing University of Information Science & Technology Nanjing 210044, China

    E-mail: [email protected], [email protected]

    Mobile number: (+86)183-6209-7758

    https://www.researchgate.net/profile/Guo-You-Zhang-2

Reviewer 2 Report

Dear Authors,

Detailed notes on the manuscript are as follows:

1) At the end of the "Introduction", clearly state the purpose of the work / research

2) 2.2. Statistical analysis - what kind of correlation did you use (Pearson, Spearman and Kendall?), What program did you use for the analysis (name and version)

3) Figure 2. Total rice planting area… TPA - the data in the chart is not linear; justify why you are using just a linear function (correlation is linear, but coefficient of determination R2 applies to every match - consider this)

4) Figure 4. Multicorrelations ... - improve the readability of the diagram (font)

5) Figure 5, 7 - as above

6) Figure 7. Rice production… what do the error bars mean; it is probably a standard error (or 5%), you should also describe the estimation and type of error in 2.2. Statistical analysis

7) Figure 8. Grain yield - as above.

8) Figure 9. Potential - as above.

Author Response

Dear reviewer

Thanks very much for your time and valuable comments on our manuscript, which would definitely improve the presentation of our study. We have revised the manuscript according to your comments, details are as follows:

  1. The introduction part was revised: study background, the topic, problem statement, and research gaps as well as the purpose of the study were clarified.
  2. Added details for the statistical work in Methodology section.
  3. Figure 2. Polynomial regression was adopted and the reason was added in figure caption.
  4. Figure 4, 5 and 7 were revised (type, color, font) to improve the presentation.
  5. Figure 7, 8 and 9 were revised more details were added.

Thanks very much for your time and comments.

On behalf of all coauthors 

Guoyou ZHANG, Ph.D, Associate Professor

School of Applied Meteorology,

Nanjing University of Information Science & Technology Nanjing 210044, China

E-mail: [email protected], [email protected]

Mobile number: (+86)183-6209-7758

https://www.researchgate.net/profile/Guo-You-Zhang-2